# Characterization of the Gut Microbiome in Healthy Dogs and Dogs with Diabetes Mellitus

**DOI:** 10.3390/ani13152479

**Published:** 2023-08-01

**Authors:** Tsz Ching Kwong, Eddie Chung Ting Chau, Mark Chi Ho Mak, Chi Tung Choy, Lee Tung Chan, Chun Keung Pang, Junwei Zhou, Phoebe Hoi Ching Poon, Yuqiong Guan, Stephen Kwok Wing Tsui, Shun Wan Chan, George Pak Heng Leung, William Chi Shing Tai, Yiu Wa Kwan

**Affiliations:** 1School of Biomedical Sciences, The Chinese University of Hong Kong, Hong Kong, China; chungtingchau@cuhk.edu.hk (E.C.T.C.); michellechan.lt@gmail.com (L.T.C.); danielpang@link.cuhk.edu.hk (C.K.P.); 1155175393@link.cuhk.edu.hk (P.H.C.P.); yuqiongjoannaguan@gmail.com (Y.G.); kwtsui@cuhk.edu.hk (S.K.W.T.); 2Non-Profit Making Veterinary Services Society Limited, Hong Kong, China; mark.mak@npv.org.hk; 3Microbiome Research Centre, BioMed Laboratory Company Limited, Hong Kong, China; zeochoy@biomed.com.hk (C.T.C.); waynezhou@biomed.com.hk (J.Z.); 4Department of Food and Health Sciences, Faculty of Science and Technology, Technological and Higher Education Institute of Hong Kong, Hong Kong, China; swchan@thei.edu.hk; 5Department of Pharmacology and Pharmacy, The University of Hong Kong, Hong Kong, China; gphleung@hku.hk; 6Department of Applied Biology and Chemical Technology, The Hong Kong Polytechnic University, Hong Kong, China; william-cs.tai@polyu.edu.hk

**Keywords:** diabetes mellitus, microbiome, canine

## Abstract

**Simple Summary:**

The gut microbiome plays a crucial role in many aspects of canine health, such as metabolism, immune function, and even behavior. The canine gut microbiome is an important and emerging field of veterinary research, with promising potential in facilitating disease diagnosis and management. This first-of-its-kind study aims to characterize the gut microbiome of dogs with diabetes mellitus in Hong Kong (compared to that of healthy controls) to shed light on its association with diabetes mellitus and its implications for disease control.

**Abstract:**

With a close pathogenetic resemblance to human diabetes, canine Diabetes Mellitus, a chronic metabolic disease featuring abnormally high blood sugar levels, is increasing in prevalence worldwide. Unlike humans, canine glycemic control requires life-long insulin injections and dietary control in most cases, thereby jeopardizing diabetic dogs’ quality of life and increasing the difficulty of disease control. While many research studies have focused on elucidating the relationship between the canine gut microbiome and diseases, there is currently no research on the subject of diabetes mellitus in dogs. We hypothesized that the gut microbiome of canines with diabetes mellitus is different from that of healthy controls. Thus, we performed targeted 16S rRNA sequencing and comprehensive bioinformatic analysis to compare the gut microbiome profiles of 16 diabetic dogs with those of 32 healthy dogs. *Clostridioides difficile*, *Phocaeicola plebeius*, *Lacrimispora indolis*, and *Butyricicoccus pullicaecorum* were found to be enriched in diabetic dogs. A distinct shift towards carbohydrate degradation metabolic pathways was found to be differentially abundant in the diabetic subjects. Alteration of the co-occurrence network was also evident in the diabetic group. In conclusion, our study suggests that the gut microbial landscape differs in diabetic canines at the genera, species, functional, and network levels. These findings have significant implications for disease management, and thus warrant further research.

## 1. Introduction

Diabetes mellitus is a relatively common endocrine disorder in dogs, with a prevalence of 23.6 cases per 10,000 dogs in the United Kingdom [1,2]. Alarmingly, there has been a notable increase in the incidence of canine diabetes mellitus in the United States, with cases rising by approximately 80% since 2006 [3,4]. Diabetic dogs typically exhibit noticeable and rapidly progressive symptoms akin to the clinical manifestation of type 1 diabetes in humans, with an extreme loss of pancreatic β-cells and insulin deficiency [5]. The typical signs of clinical diabetes mellitus due to persistent hyperglycemia are increased hunger (polyphagia, PP) and thirst (polydipsia, PD), frequent urination (polyuria, PU), and weight loss [6,7]. While serum fructosamine measurement is the most commonly used diagnostic tool for assessing canine diabetes mellitus, glycosylated hemoglobin A1c (HbA1c) tests and the measurement of glucose levels in urine and blood can also be used for diagnosing the disease and monitoring its progress [8,9,10]. In addition, the level of ketone bodies in urine should be monitored, as diabetic ketoacidosis occurs in dogs with poorly controlled diabetes mellitus [6]. Thus far, there is no readily available, standardized laboratory test with which to identify the underlying cause of diabetes mellitus in dogs, which delays its diagnosis until the late stages of the disease [11].

Canine diabetes mellitus management is often intractable due to rapid and progressive changes in a dog’s condition, and fluctuating therapeutic responses [12]. Current treatments for canine diabetes mellitus mainly rely on reducing carbohydrates and exogenous insulin treatment to maintain blood glucose levels under the renal threshold for the greatest possible duration within a 24 h timeframe [6]. Non-insulin therapeutic agents are rarely (if ever) sufficient for use as sole anti-diabetic therapeutics for dogs. Currently, although more scientific evidence is needed to enable widespread clinical use, promising results have been reported regarding diabetic dogs treated with incretin-based therapeutics such as Liraglutide [13,14].

The canine gut microbiome is an important and emerging field of veterinary research. The gastrointestinal tract contains a complex microbial community, and the dysbiosis of such microbiomes leads to several diseases and disorders [15]. Knowing the diversity and taxonomic bacterial distribution of the gut microbiota may provide hints regarding the onset and progression of corresponding diseases. Accumulating evidence suggests that various diseases, including cancers and cardiovascular diseases, affect canine gut microbial composition [16,17,18,19]. Yet, no such research has been conducted regarding canine diabetes mellitus. The objective of this study was to identify the characteristics of the gut microbiota in non-diabetic and diabetic dogs sharing comparable lifestyles and living conditions in Hong Kong via targeted 16S rRNA sequencing. Using statistical and bioinformatics approaches, the microbial landscape of the canine gut microbiome in relation to diabetes mellitus was highlighted at various levels.

## 2. Materials and Methods

### 2.1. Experimental Design and Study Population

A total of 32 healthy dogs and 16 dogs diagnosed with diabetes mellitus from the Non-Profit Making Veterinary Services Society Animal Hospital (NPVAH) in Hong Kong were enrolled in the present study, and informed consent was obtained from their owners. Breed size was classified by the mean weight provided by the American Kennel Club (AKC). The age group determination method was adopted from the Canine Life Stage Guidelines 2019 issued by the American Animal Hospital Association (AAHA) [20]. Healthy dogs were selected based on the results of their veterinary check-ups, which consisted of the following information: absence of signs of sufferance, inflammation, gastrointestinal problems, and chronic conditions. Dogs diagnosed with diabetes mellitus were selected based on a diagnosis by a veterinarian and clinical history. Samples of dogs with diabetes mellitus, regardless of treatment status, were chosen for the diabetic group, and samples from the healthy group were used as controls.

### 2.2. Sample Collection and 16S rRNA Sequencing

Rectal or stool samples were collected with sterile swabs from dogs diagnosed with diabetes mellitus and healthy dogs at NPVAH, and promptly stored in a 2 mL preservation buffer at 4 °C [21]. All samples were transported to the laboratory within 7 days and processed within 3 days. Microbial genomic DNA was extracted using a QIAamp PowerFecal Pro DNA Kit (Qiagen, Hilden, Germany). The 16S rRNA sequences were analyzed as described previously [22]. Briefly, a Nextera XT DNA Library Preparation Kit was used for the amplification of the V3-V4 region in the 16S rRNA gene. Paired-end sequencing was carried out on the NovaSeq^TM^ platform (Illumina, San Diego, CA, USA) by Novogene (Hong Kong, China). Prior to downstream analysis, index barcodes and adapter sequences were trimmed from pair-ended demultiplexed reads.

### 2.3. Sequencing Data and Bioinformatics Analysis

Sequencing data were analyzed using the Quantitative Insights into Microbial Ecology (QIIME) 2-2023.2 [23]. Demultiplexed reads were quality controlled and denoised with DADA2 [24] to retrieve exact amplicon sequence variants (ASVs). All ASVs were then aligned by MAFFT [25], and then a phylogenetic tree was generated using fastree2 [26] via the q2-phylogeny plugin. The taxonomic annotation of the resulting ASV was carried out using the q2-feature-classifier plugin [27] and a pre-trained Naive Bayes classifier which was based on the SILVA v138 taxonomic reference database with 99% similarity [28,29]. We used six metrics to indicate alpha diversity: the Observed OTUs, Chao1 Index (Chao1), ACE Index (ACE), Shannon Diversity Index (Shannon), Simpson Index (Simpson), and Faith’s phylogenetic diversity (PD). In addition, beta diversity was calculated based on the Jaccard distance metric, Bray–Curtis distance metric, weighted UniFrac, and unweighted UniFrac distance metrics. The PERMANOVA test on beta diversity (999 permutations) was applied to compare the microbial community dissimilarity across groups [30]. Adonis was applied to investigate microbial community dissimilarity across gender, age group, breed size, and neutered status [30]. Differential abundance analysis was conducted by ANCOM with bias correction (ANCOM-BC) [31]. The co-occurrence/co-exclusion network was inferred by the Sparse and Compositionally Robust Inference of Microbial Ecological Networks (SPIEC-EASI) framework using the neighborhood selection framework introduced by Meinshausen and Bühlmann [32,33].

### 2.4. Statistical Analysis

All the statistical analysis and visualization of results were conducted in Python 3.9.13 (numpy version 1.23.5, scipy version 1.10.1, matplotlib version 3.7.1 and seaborn version 0.12.1) and Flourish (https://flourish.studio, accessed on 1 June 2023). Normality assumptions were evaluated using D’Agostino and Pearson’s test (scipy.stats.normaltest function), and the Shapiro–Wilk test (scipy.stats.shapiro function) if parametric tests were employed. Demographic characteristics were evaluated using the non-parametric Mann–Whitney U rank test (scipy.stats.mannwhitneyu function) for continuous variables, and the Fisher exact test for categorical variables (scipy.stats.fisher_exact function). *p*-value correction was performed with the statsmodels.stats.multitest.multipletests function using a Benjamini/Hochberg (non-negative) procedure. All values were expressed as mean ± standard error of the mean (SEM), and *p* < 0.05 was considered statistically significant unless otherwise specified.

## 3. Results

### 3.1. Demographics of Study Population

Our study population (*n* = 48) contained 16 different canine breeds with approximately 80% of the population small in breed size and neutered. The Toy Poodle (*n* = 21) and mixed breed mongrels (*n* = 6) were the most common breeds. The demographics and disease characteristics of the recruited subjects was summarized in Table 1. There was no statistically significant difference observed in mean body weight (*p* > 0.05, Mann–Whitney U), gender, breed size, and neutered status (*p* > 0.05, Chi-square test) between diabetic and healthy groups. There was a statistically significant discrepancy in mean age (*p* < 0.0001, Mann–Whitney U) and age group (*p* < 0.05, Chi-square test) between the two groups; thus, age group adjustment was applied in subsequent analysis unless otherwise specified.

### 3.2. Significant Differences in Microbial Diversity between Diabetic and Healthy Dogs

No significant difference was observed between the two groups, irrespective of age group, in terms of ACE, Chao1, Faith’s PD, Observed OTUs, Shannon, or Simpson, as shown in Appendix A. Owing to the difference in age distribution mentioned in Section 3.1, we performed subgroup analysis to stratify the samples by age group. A significant difference in alpha diversity by Faith’s PD was revealed between diabetic and healthy dogs in both adults and senior subgroups, respectively (Figure 1).

The beta diversity of diabetic subjects was significantly different from that in the healthy group in terms of Bray–Curtis (*p* = 0.027, PERMANOVA) and Jaccard distance (*p* = 0.002, PERMANOVA), which was demonstrated by the distinctive clustering in the principal coordinates (PCoA) analysis biplot (Figure 2A,B). Age group discrepancy was observed in Bray–Curtis, Cosine, and Jaccard, as reflected by the Adonis test; however, additional statistical significance was observed between groups in the unweighted UniFrac distance matrix (Appendix A). For the Firmicutes/Bacteroidetes ratio, no significant difference was observed between the two groups, as shown in Figure 2C (*p* = 0.702, Mann–Whitney U).

A total of 8883 unique amplicon sequence variants (ASV) were identified, of which 312 ASVs were categorized as rare ASVs, with exactly one count through the dataset (Figure 2E,F). After alignment, these ASVs were assigned to 47 phyla, 1059 genera, and 2019 species. The most abundant phyla were Proteobacteria and Firmicutes (Figure 2D), including significantly enriched genera such as *Sphingomonas*, *Ralstonia*, *Pseudomonas*, *Succinivibrio*, *Bacillus*, *Saccharofermentans, Achromobacter, Methyloversatilis and Sphingobium* (*p* < 0.05, Mann–Whitney U). 

### 3.3. Differential Abundance between Healthy and Diabetic Groups

Differential abundance at the ASV level was analyzed via ANCOM-BC with age group adjustment, as shown in Figure 3A and Table 2. A total of four ASVs from four distinct genera (*Clostridioides*, *Bacteroides*, *Anaerostipes,* and *Butyricicoccus)* were found to be differentially expressed in the diabetic canines (adjusted for age group, gender, and breed size). In the diabetic group, *Clostridioides difficile* was the most differentially abundant species, with a 3-fold increase found, while a high percentage of *Phocaeicola plebeius*, *Lacrimispora indolis,* and *Butyricicoccus pullicaecorum* was also observed, with at least a 2-fold increment when compared to the healthy group (Figure 3B–E) (*p* < 0.0001, Mann–Whitney U).

### 3.4. Enhanced Carbohydrate-Related Degradation of Functional Abundance in the Diabetic Group

To more precisely reflect the physiological consequence of the gut microbiome profile, functional abundance was deduced via PICRUSt2 following LefSe. A total of 44 distinct features were identified with a cut-off of log-linear discriminant analysis (LDA) score at 2 (Figure 4A,B). Most of the features over-represented in the control group were related to the synthesis of essential molecules and cell structure components, such as L-methionine, L-glutamine, mannose, and nucleotides (guanosine, purine, pyrimidine). In addition, pathways of central microbial metabolism including aerobic respiration, pyruvate fermentation, and photorespiration were enriched in the control group. Conversely, nine MetaCyc functional pathways involved in the degradation of essential bio-elements, mainly carbohydrates, were upregulated in diabetic dogs, as detailed in Table 3. The most abundant pathway, PWY-621 (sucrose degradation), was elevated in the diabetic group (log LDA = 2.919, *p* = 0.002), which is a microbial metabolism responsible for the conversion of sucrose into hexose as the energy source for growth. This may reveal the potential functional role of gut microbiota in alleviating hyperglycemia in diabetic dogs, which will be elaborated upon further in the Section 4

### 3.5. Remodeling of the Microbial Co-Occurrence/Exclusion Network

The microbial co-occurrence/exclusion network among the two groups, comprising 561 nodes representing each amplicon sequence variant (ASV), was determined via SPIEC-EASI. The total number of edges was 3733 and 3163 in the diabetic and healthy groups, respectively. For better visualization, the topology of nodes was fixed across the groups, with positive (co-occurrence) or negative (co-exclusion) edges between connecting nodes denoted by blue and red lines, respectively. 

In brief, a distinct edge distribution was observed in the diabetic group, which was remodeled in the healthy group (Figure 5A,B). The most co-occurrence relationships (i.e., four distinct hub notes of degree >100) in the diabetic dogs derived from the *Sodalis*, *Helicobacter,* and *Alloprevotella* genera (Appendix A). Concerning phylogeny, three out of the top ten hub nodes were shared across all groups: *Lachnospiraceae* family, *Bacteroides,* and *Helicobacter* genera (Figure 5C). The difference in networks was further supported by centrality measures, including betweenness, closeness, degree, and eigenvector, as shown in Figure 5D.

## 4. Discussion

Research on both human and canine studies has shown that changes in the fecal microbiota can lead to gastrointestinal dysbiosis, which can be influenced by diseases [15]. To better understand the link between specific diseases and gastrointestinal dysbiosis in dogs, it is important to identify distinct patterns in their gut microbiome. Canine diabetes mellitus, a chronic metabolic disease that is difficult to manage, is common in middle-aged and older dogs. It can impair the entire endocrine system, thereby affecting carbohydrate, protein, and lipid metabolism with rapid progression [34]. Failure to manage canine diabetes mellitus may lead to fatal outcomes due to various metabolic complications. To the best of our knowledge, this is the first gut microbiome study on diabetic dogs aiming to clarify differences in the gut microbiome profile, in particular at the functional and network levels. 

In the present study, clear differences in the gut microbiota of diabetic and healthy dogs were demonstrated. This was evident in the distinct clustering of the two groups in the PCoA plots based on Bray–Curtis and Jaccard distances, as well as the differential abundance analysis at the genera level. The phyla *Actinobacteria*, *Bacteroidetes*, *Firmicutes*, *Fusobacteria,* and *Proteobacteria* were detected in both groups, aligning with prior research on the gut microbiome of canines [16,19,35]. We observed that the family *Clostridiaceae* was overexpressed, consistent with previous studies on insulin-dependent diabetic dogs [36,37]. However, the family *Enterobacteriaceae* was not significantly enriched in our study population. This disparity may be attributed to breed variations or age-related differences in the gut microbiome, rendering direct comparison with previous research inappropriate.

Prominently in diabetic dogs, there was a significant increase in the presence of *Clostridioides* (previously known as *Clostridium*) *difficile*, an opportunistic pathogen that can cause severe gastrointestinal discomforts. *C. difficile* infection (CDI) is a major cause of hospital-associated and antibiotic-associated diseases in humans [38,39] and can lead to symptoms ranging from diarrhea to life-threatening damage to the colon [40]. Studies have shown that chronic diarrhea is a common intestinal manifestation of diabetes mellitus in dogs [41], while it remains a risk factor for recurrent CDI in humans [40]. However, the causal relationship between CDI and canine enteric diseases has not yet been fully elucidated [42], although there are indications that it could be a possible causative agent of chronic diarrhea [42,43,44,45]. Additionally, it is important to note that zoonotic transmission of *C. difficile* between pets and their owners can occur due to frequent interactions (e.g., licking and petting) and sharing of living environments [46]. As a result, veterinarians should consider gut dysbiosis in the clinical management of diabetic dogs, and gut microbiome testing may serve as a useful tool to detect increased susceptibility to CDI in canines with diabetes mellitus. 

After subsequent functional pathway and network analysis in the gut microbiome, we found a possible reason for the difference observed. At least 50% of diabetic dogs resemble the clinical feature of inadequate insulin production in Type I diabetes in humans [11]. Thus, the lack of insulin causes issues with breaking down carbohydrates, which is evident in the higher levels of functional traits related to carbohydrate degradation pathways observed in this study. This could be due to the excessive growth of microorganisms that utilize carbohydrates as their primary energy source in response to the carbohydrate-rich gut environment. Coincidently, it is well documented in humans that primary metabolites produced by microbial carbohydrate degradation, especially short-chain fatty acids (SCFAs), help maintain intestinal homeostasis and have anti-inflammatory effects in patients with metabolic diseases and diabetes mellitus [47,48]. Nevertheless, our findings suggest a possible mechanism through which the gut microbiome affects diabetic dogs; yet, further research is needed to understand it fully.

Analysis of the in silico pathway and co-occurrence/-exclusion networks indicated that canines with diabetes mellitus have microbial dysbiosis in their gut. The network connection patterns changed dynamically, with four hub nodes emerging in this study. These changes indicate that the gut microenvironment of dogs with diabetes mellitus is undergoing remodeling. Although the exact mechanism of such remodeling remains unknown, it is speculated that heterotrophic bacteria such as *Lacrimispora indolis*, *Phocaeicola plebeius,* and *Butyricicoccus pullicaecorum* observed in this study are capable of adapting their metabolic capabilities to survive better in the carbohydrate-rich gut environment of dogs. 

Canine diabetes mellitus is more prevalent among certain dog breeds, particularly those that are smaller in size, such as Poodles (Toy/Miniature) [34,49], Pomeranians, and Terriers [34]. In densely populated cities like Hong Kong, it is unsurprising that most of the local domestic dogs are small breeds, as our study population shows. Notably, the overwhelming majority of our diabetic subjects (12 out of 16) were Toy Poodles. Therefore, the findings of this study provide valuable insights for dog owners in the local area, and in other regions with similarly constrained environments. 

In light of all findings presented in this study, it is evident that diabetic dogs experience changes in their gut microbiota at the taxonomical, pathway, and network levels. A notable increase in the abundance of *C. difficile* may be related to the gastrointestinal complications observed in canine diabetes mellitus, though further investigations are needed to characterize the underlying mechanisms involved. The breakdown of excess carbohydrates in the gut is crucial in managing hyperglycemia in dogs with diabetes mellitus, highlighting the importance of gut microbiota. However, the small sample size and uneven age group distribution in this study may confine its statistical power due to resource constraints. Therefore, a more comprehensive longitudinal study with a larger study population is warranted. Overall, our study offers novel insights into the development of microbiome-targeted diagnostics and therapies, and provides a scientific basis for utilizing the gut microbiome in the clinical management of canine diabetes mellitus.

## 5. Conclusions

This study compared the gut microbiome of diabetic and healthy dogs living in Hong Kong with similar lifestyles and living conditions. The results showed significant differences in the prevalence of microbial species between the two groups, with *Clostridioides difficile*, *Phocaeicola plebeius*, *Lacrimispora indolis,* and *Butyricicoccus pullicaecorum* being enriched in dogs with Diabetes mellitus. A distinct shift towards carbohydrate degradation metabolic pathways was predicted to be differentially abundant in diabetic dogs, likely due to their carbohydrate-rich gut environments. Functional pathway remodeling and co-occurrence network changes were evident in this study, offering insight into the role of gut microbiota in diabetic development. In conclusion, our study was the first attempt to identify the microbial landscape at various levels involved in canine diabetes mellitus, which may aid the development of new microbiome-targeted tools to counter this disease.

## Figures and Tables

**Figure 1 animals-13-02479-f001:**
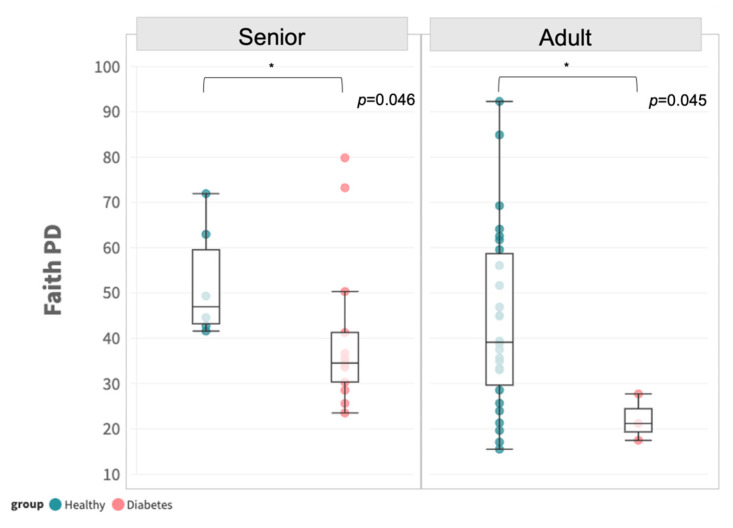
Alpha diversity in terms of Faith’s PD between adult and senior subgroups in healthy and diabetic dogs. Statistical significance of *p* < 0.05 is indicated with *.

**Figure 2 animals-13-02479-f002:**
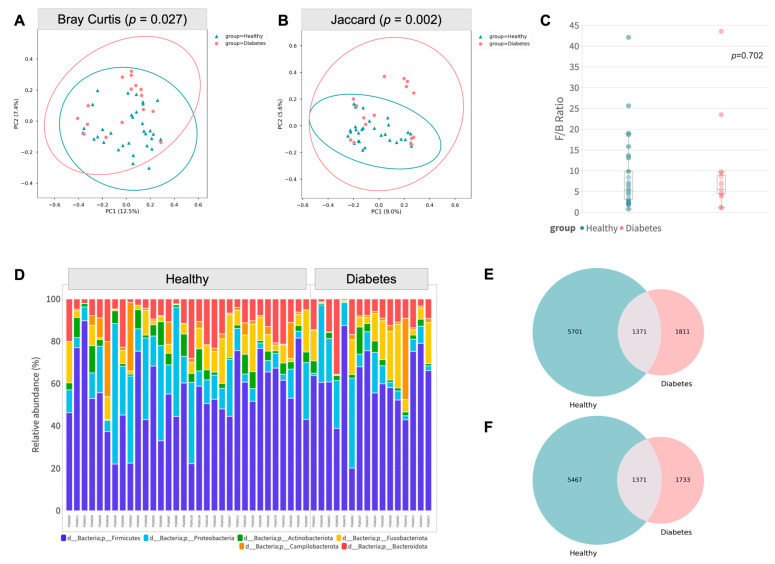
Gut composition profile of diabetic subjects. (**A**,**B**) Principal coordinate analysis biplot based on Bray–Curtis and Jaccard distances (PERMANOVA, 999 permutations). (**C**) Boxplot of Firmicutes/Bacteroidetes (F/B) ratio (Mann–Whitney U test). (**D**) Relative abundance of the dominant phyla (*Firmicutes*, *Proteobacteria*, *Actinobacteriora*, *Fusobacteriota*, *Campilobacterora,* and *Bacteroidota*) between healthy and diabetic groups. (**E**) Venn diagram of all ASV between healthy and diabetic groups. (**F**) Venn diagram of ASV (excluding rare ASV) between healthy and diabetic groups.

**Figure 3 animals-13-02479-f003:**
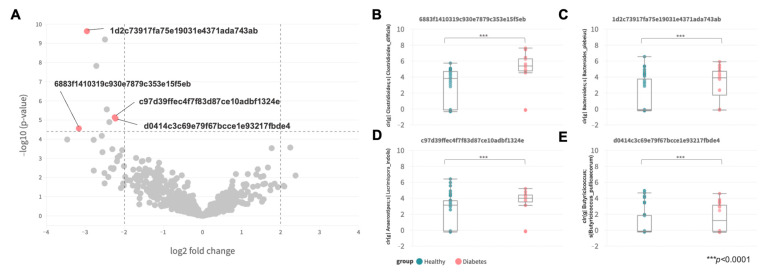
Differentially abundant ASV. (**A**) Differential abundance volcano plot of ASVs. (**B**–**E**) Boxplots of the center-log-ratio (clr)-transformed abundance of *Clostridioides difficile* (6883f1410319c930e7879c353e15f5eb)*, Phocaeicola plebeius* (1d2c73917fa75e19031e4371ada743ab)*, Lacrimispora indolis* (c97d39ffec4f7f83d87ce10adbf1324e), and *Butyricicoccus pullicaecorum* (d0414c3c69e79f67bcce1e93217fbde4), respectively.

**Figure 4 animals-13-02479-f004:**
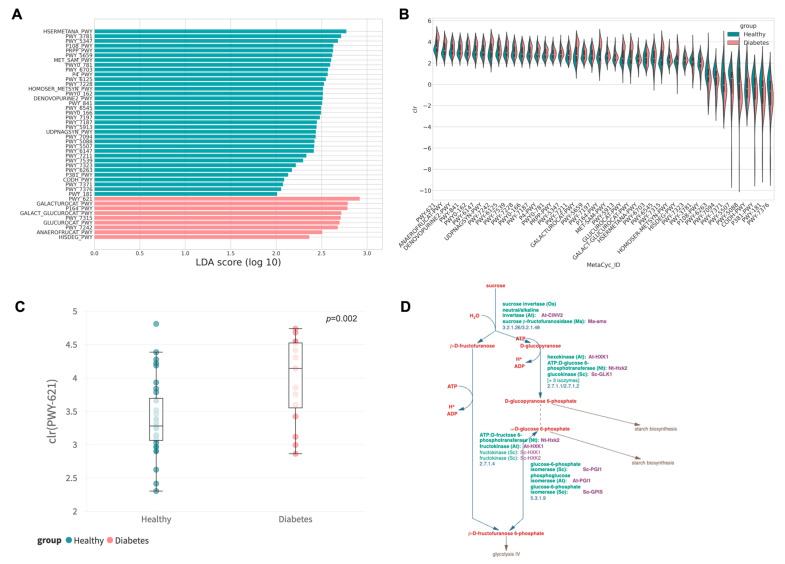
Predicted MetaCyc pathway abundance. (Green: Healthy, Pink: Diabetes) (**A**) Log LDA score of differentially abundant MetaCyc pathways. A higher Log LDA score indicates a larger effect size. (**B**) The violin plot of center-log-ratio (clr)-transformed abundance of the differentially abundant MetaCyc pathways identified via LefSe. (**C**) Boxplot of center-log-ratio (clr)-transformed abundance of PWY-621. (**D**) Pathway diagram of PWY-621 from MetaCyc, indicating the detailed microbial metabolism of sucrose degradation into hexose for energy expenditure.

**Figure 5 animals-13-02479-f005:**
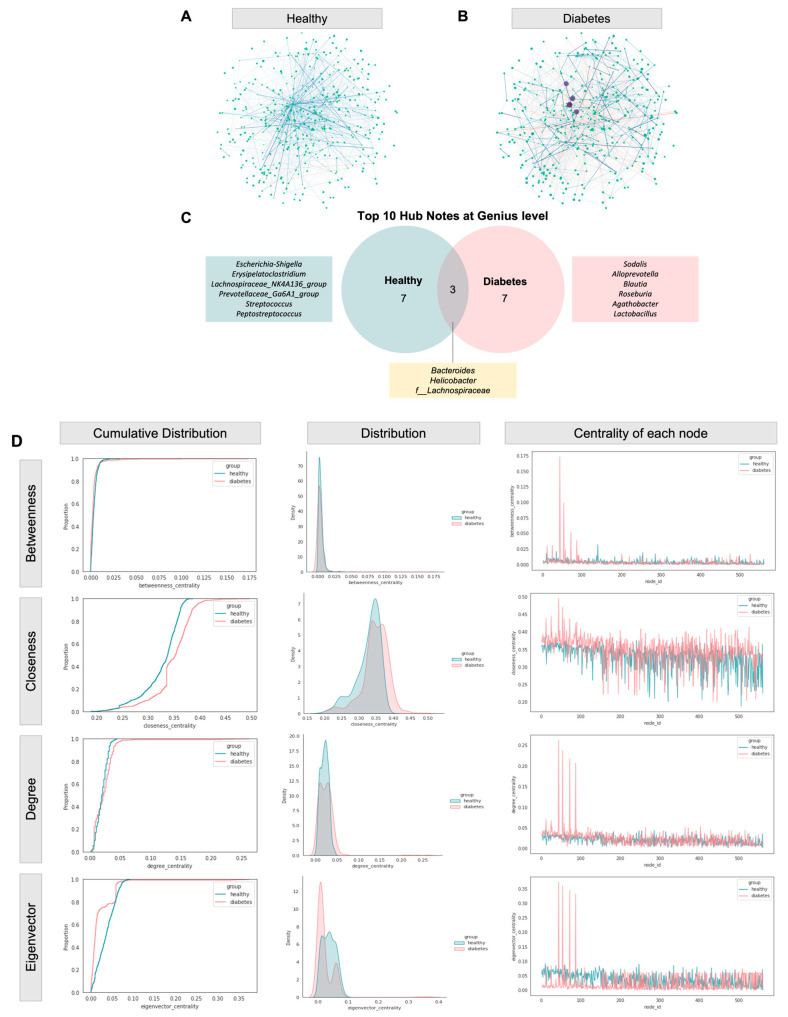
Co-occurrence/co-exclusion network. (**A**,**B**) Co-occurrence network diagram of diabetic and healthy groups. The node size is proportional to the respective number of degrees, while the edge width is proportional to the strength of association. A blue and red labeled edge represents a positive and negative association, respectively. (**C**) Top ten hub nodes at genus level across networks. Co-occurrence/co-exclusion network. (**D**) Empirical cumulative distribution function (eCDF) (**left**) and kernel density distribution (**middle**) and centrality of each node (**right**) of betweenness, closeness, degree, and eigenvector centrality measures.

**Table 1 animals-13-02479-t001:** Demographics and disease characteristics of the study population.

	Diabetes Mellitus (*n* = 16)	Healthy (*n* = 32)	*p*
Mean body weight (kg)	6.10 ± 0.71	9.86 ± 1.67	0.59
Mean age (years)	10.82 ± 0.89	5.28 ± 0.56	<0.0001
Gender, No. (%)			
Male	5 (31.25)	14 (43.75)	0.40
Female	11 (68.75)	18 (56.25)	
Age group (%) ^1^			
Adult	3 (18.75)	26 (81.25)	<0.05
Senior	13 (81.25)	6 (18.75)	
Breed size (%) ^2^			
Small	14 (87.50)	25 (78.12)	0.43
Medium	2 (12.50)	7 (21.88)	
Neutered status (%)			
Yes	13 (81.25)	24 (75.00)	0.63
No	3 (18.75)	8 (25.00)	

^1^ Mean weight less than 14.0 kg is classified as Small, and between 14.0 and 25.0 kg is classified as Medium. ^2^ Age group is determined according to the breed size and age: Small—Adult: between 1.0 and 11.0 year-old, Senior: > 11.0 years-old; Medium—Adult: between 1.0 and 8.0 year-old, Senior: > 8.0 year-old.

**Table 2 animals-13-02479-t002:** Differentially abundant ASVs (taxonomic unit assigned by a q2-feature-classifier) identified via ANCOM-BC.

Feature ID	Taxon	*p*
6883f1410319c930e7879c353e15f5eb	f__Peptostreptococcaceae; g__Clostridioides; s__Clostridioides_difficile	<0.0001
1d2c73917fa75e19031e4371ada743ab	f__Bacteroidaceae; g__Bacteroides; s__Phocaeicola_plebeius	<0.0001
c97d39ffec4f7f83d87ce10adbf1324e	f__Lachnospiraceae; g__Anaerostipes; s__Lacrimispora_indolis	<0.0001
d0414c3c69e79f67bcce1e93217fbde4	f__Butyricicoccaceae; g__Butyricicoccus; s__Butyricicoccus_pullicaecorum	<0.0001

**Table 3 animals-13-02479-t003:** Differentially abundant MetaCyc pathways deduced by PICRUSt2 and LefSe.

BioCyc ID	MetaCyc Pathway Name	Group	Log LDA	*p*
HSERMETANA-PWY	L-methionine biosynthesis III	Healthy	2.772	0.001
PWY-3781	aerobic respiration I (cytochrome c)	Healthy	2.715	0.017
PWY-5347	superpathway of L-methionine biosynthesis (transsulfuration)	Healthy	2.680	0.004
P108-PWY	pyruvate fermentation to propanoate I	Healthy	2.629	0.003
PRPP-PWY	superpathway of histidine, purine, and pyrimidine biosynthesis	Healthy	2.625	0.016
PWY-5659	GDP-mannose biosynthesis	Healthy	2.618	0.001
MET-SAM-PWY	superpathway of *S*-adenosyl-L-methionine biosynthesis	Healthy	2.605	0.019
PWY0-781	aspartate superpathway	Healthy	2.595	0.002
PWY-6703	preQ_0_ biosynthesis	Healthy	2.571	0.005
P4-PWY	superpathway of L-lysine, L-threonine and L-methionine biosynthesis I	Healthy	2.567	<0.001
PWY-6125	superpathway of guanosine nucleotides de novo biosynthesis II	Healthy	2.547	0.034
PWY-7228	superpathway of guanosine nucleotides de novo biosynthesis I	Healthy	2.527	0.049
HOMOSER-METSYN-PWY	L-methionine biosynthesis I	Healthy	2.516	0.018
PWY0-162	superpathway of pyrimidine ribonucleotides de novo biosynthesis	Healthy	2.514	0.042
DENOVOPURINE2-PWY	superpathway of purine nucleotides de novo biosynthesis II	Healthy	2.512	0.030
PWY-841	superpathway of purine nucleotides de novo biosynthesis I	Healthy	2.507	0.030
PWY-6545	pyrimidine deoxyribonucleotides de novo biosynthesis III	Healthy	2.494	0.027
PWY0-166	superpathway of pyrimidine deoxyribonucleotides de novo biosynthesis (*E. coli*)	Healthy	2.494	0.011
PWY-7197	pyrimidine deoxyribonucleotide phosphorylation	Healthy	2.481	0.049
PWY-7187	pyrimidine deoxyribonucleotides de novo biosynthesis II	Healthy	2.448	0.014
PWY-5913	partial TCA cycle (obligate autotrophs)	Healthy	2.445	0.038
UDPNAGSYN-PWY	UDP-*N*-acetyl-D-glucosamine biosynthesis I	Healthy	2.437	0.044
PWY-7094	fatty acid salvage	Healthy	2.435	0.011
PWY-5088	L-glutamate degradation VIII (to propanoate)	Healthy	2.422	0.001
PWY-5507	adenosylcobalamin biosynthesis I (anaerobic)	Healthy	2.418	0.003
PWY-6147	6-hydroxymethyl-dihydropterin diphosphate biosynthesis I	Healthy	2.415	0.034
PWY-7211	superpathway of pyrimidine deoxyribonucleotides de novo biosynthesis	Healthy	2.332	0.042
PWY-7539	6-hydroxymethyl-dihydropterin diphosphate biosynthesis III (Chlamydia)	Healthy	2.297	0.044
PWY-7323	superpathway of GDP-mannose-derived O-antigen building blocks biosynthesis	Healthy	2.217	0.049
PWY-6263	superpathway of menaquinol-8 biosynthesis II	Healthy	2.173	0.001
P381-PWY	adenosylcobalamin biosynthesis II (aerobic)	Healthy	2.131	0.007
CODH-PWY	reductive acetyl coenzyme A pathway I (homoacetogenic bacteria)	Healthy	2.087	0.049
PWY-7371	1,4-dihydroxy-6-naphthoate biosynthesis II	Healthy	2.075	0.001
PWY-7376	cob(II)yrinate *a,c*-diamide biosynthesis II (late cobalt incorporation)	Healthy	2.055	0.010
PWY-181	photorespiration I	Healthy	2.010	0.042
PWY-621	sucrose degradation III (sucrose invertase)	Diabetes	2.919	0.002
GALACTUROCAT-PWY	D-galacturonate degradation I	Diabetes	2.788	0.003
P164-PWY	purine nucleobases degradation I (anaerobic)	Diabetes	2.784	0.024
GALACT-GLUCUROCAT-PWY	superpathway of hexuronide and hexuronate degradation	Diabetes	2.717	0.002
PWY-7315	dTDP-*N*-acetylthomosamine biosynthesis	Diabetes	2.709	0.007
GLUCUROCAT-PWY	superpathway of β-D-glucuronosides degradation	Diabetes	2.698	<0.001
PWY-7242	D-fructuronate degradation	Diabetes	2.678	0.009
ANAEROFRUCAT-PWY	homolactic fermentation	Diabetes	2.506	0.029
HISDEG-PWY	L-histidine degradation I	Diabetes	2.364	0.044

## Data Availability

Data will be available on request from the corresponding authors.

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
