# Peer review of "Characterization of the Gut Microbiome in Healthy Dogs and Dogs with Diabetes Mellitus"

_animals, 2023, doi:10.3390/ani13152479_

Round 1

Reviewer 1 Report

*Phrasing starting on line 43 is cumbersome and needs editing.
*Line 55 - is adjourn the correct word?

Paper seems like it was written by two different people - Conclusions section is much more readable/better quality of English.

Author Response

Thank you for your valuable comments, we’ve revised the manuscript accordingly. Please find the point-to-point response in blue as follow:

*Phrasing starting on line 43 is cumbersome and needs editing.

The sentence has been rephrased as “Alarmingly, there has been a notable increase in the incidence of canine Diabetes Mellitus in the United States, with cases rising by approximately 80% since 2006 [3,4].”

*Line 55 - is adjourn the correct word?

“adjourn” has been replaced by “delay”

Reviewer 2 Report

This article begins with an overview of DM prevalence in the canine population, its diagnosis, and difficulties related to management. The objective of the study was to compare the fecal microbiota of diabetic dogs to non-diabetic controls, interpreted in the light of the diabetic disease state. 16s rRNA sequencing was used to evaluate the bacterial population. Phylogenetic diversity was slightly statistically different between the groups and, as was beta diversity and differential abundance. Several differences in bacterial distribution were identified between the groups. The discussion covers some of the findings including C. difficile appearing to be higher in prevalence in diabetic dogs and how some of the bacterial species may be playing roles in response to the carbohydrate-rich environment.

The abstract of the study should be amended to reflect the hypothesis and findings more clearly as discussed in the specific comments. There is no specific hypothesis identified which needs to be done before choosing a tailed T-test. The objective was to describe the characteristics, so a hypothesis that there would be a difference, or that carbohydrate metabolism would be different, upregulated, etc. 

Overall the study design was appropriate with cases and controls other than the concern raised regarding age differences and lack of determination of controlling for diet, obesity, diabetic control, antibiotic administration and other potential relevant factors. 

Some of the particular weaknesses of this study include the limited population group (small/toy breeds in Hong Kong) and the age difference between the diabetic and control groups. It is also recognized that faecal microbiome distribution can vary over time, so controlling for intraindividual temporal variation with repeated measurements would be ideal. In addition, discussion of if the power of the study was adequate to identify these statistical differences would be appropriate. 

While this study appears to be the first evaluating functional and network differences between diabetic and healthy dogs, several others have evaluated the taxonomic distribution: Laia et al (PLoS One 2022 doi: 10.1371/journal.pone.0273792) and Kim et al (In Vivo 2023 doi: 10.21873/invivo.13130) have evaluated longitudinal microbiome distributions in diabetic dogs, and Jergens et al (Frontiers in Veterinary Science 2019 doi: 10.3389/fvets.2019.00199), so discussion of these results in comparison to this study’s is appropriate.

Overall this paper presents novel information and would add to the special report on microbial genomics, particularly the implications for microbiome compensation to different metabolic states.

Lines 30-35: Within the abstract, it is not clear what the hypothesis versus findings were - Lines 30-32 describe the methods, lines 32-33 describe a result, then 33-34 describe a predicted outcome, and lines 34-35 go back to results. The hypothesis should be more clearly defined and placed before the results, not in between. Alternatively, it may be that “pathways were predicted to be differentially abundant”  is not the correct phrase and in fact it should be”pathways were found to be differentially abundant.”

Line 50: HbA1c is noted as a diagnostic tool, but more commonly fructosamine testing is used to confirm persistent hyperglcyemia in canines so if a diagnostic review is discussed, it should include serum fructosamine measurement. 

Line 85-90: Were the dogs reviewed for diet history, recent antibiotic use, obesity, diabetic control, or other variables that might impact the gut microbiome?

Line 133-138: Typically the study population is included in the Materials & Methods section with a notation of the statistically relevant differences in results. 

Line 159-161: The age group discrepancy should be paired with the other results regarding age distribution rather than in between two sections discussing diabetic vs. control populations. 

Figure 2A & 2B: the keys for the principal coordinate analyses plots should be moved so the symbols are not being overlapped and hidden by the coordinates. 

Lines 172-175:: Brief description of what the diagrams and graphs demonstrate could be included in the figure description

Line 202: The term “ameliorated” here does not seem to fit or describe the results well - ameliorate means to improve or make better, and I suspect what was actually meant was “upregulated” or “showed enhanced activity” and should be reworded to reflect the results more accurately.

Figure 4A: Key needed

There are several areas where grammatical and sentence construction editing are required. There are a few specific areas where word choice clouds comprehension significantly - these have been denoted in the specific comments. 

Author Response

Thank you for your valuable comments and pointing out the weaknesses of this study. We agreed that repeated measurements to account for intraindividual temporal variation would be insightful as changes in faecal microbiome might occur with disease progression. Given to the constraints of resources, we added these limitations to the discussion section (lines 327 - 329) and will consider your suggestion in further studies. Also, we have included the comparison to previous studies on the taxonomic distribution in diabetics dogs in lines 268 - 273 as you suggested. We’ve revised the manuscript according to your comments and please find the point-to-point response in blue as follow: 

Lines 30-35: Within the abstract, it is not clear what the hypothesis versus findings were - Lines 30-32 describe the methods, lines 32-33 describe a result, then 33-34 describe a predicted outcome, and lines 34-35 go back to results. The hypothesis should be more clearly defined and placed before the results, not in between. Alternatively, it may be that “pathways were predicted to be differentially abundant”  is not the correct phrase and in fact it should be”pathways were found to be differentially abundant.”

We added the sentence “We hypothesized that the gut microbiome of canine Diabetes Mellitus is different from that of healthy controls.” in line 29 before discussing the results. Also, line 34 has been modified to ”pathways were found to be differentially abundant.”

Line 50: HbA1c is noted as a diagnostic tool, but more commonly fructosamine testing is used to confirm persistent hyperglcyemia in canines so if a diagnostic review is discussed, it should include serum fructosamine measurement. 

We rephrased line 50 and incorporated your suggestion as follows: “While serum fructosamine measurement is the most commonly used diagnostic tool in canine Diabetes Mellitus, glycosylated haemoglobin A1c (HbA1c) tests and measurement of glucose level in urine and blood can also be used for diagnosis and monitoring the progress of canine DM [8–10].”

Line 85-90: Were the dogs reviewed for diet history, recent antibiotic use, obesity, diabetic control, or other variables that might impact the gut microbiome?

We agreed these variables might impact the gut microbiome and we have included this limitation in the discussion (line 327 - 329). Since this was an observational study, we did not have access to the information of diet history and diabetic control. For obesity, we have included the mean body weight in the revision and no statistical significance was found between healthy and diabetic groups (Table 1).

Line 133-138: Typically the study population is included in the Materials & Methods section with a notation of the statistically relevant differences in results. 

We have modified the Materials & Methods section to specify the study population (line 82 - 92), while the demographics of the study population was discussed in Results section.

Line 159-161: The age group discrepancy should be paired with the other results regarding age distribution rather than in between two sections discussing diabetic vs. control populations. 

Detailed description was added to explain the rationale and results of subgroup analysis of the age distribution (lines 154 - 155).

Figure 2A & 2B: the keys for the principal coordinate analyses plots should be moved so the symbols are not being overlapped and hidden by the coordinates. 

The subfigures have been re-plotted accordingly.

Lines 172-175:: Brief description of what the diagrams and graphs demonstrate could be included in the figure description

Brief description of the figures has been added in the figure description.

Line 202: The term “ameliorated” here does not seem to fit or describe the results well - ameliorate means to improve or make better, and I suspect what was actually meant was “upregulated” or “showed enhanced activity” and should be reworded to reflect the results more accurately.

The term “ameliorated” has been changed to “upregulated” as you suggested to reflect the results more precisely.

Figure 4A: Key needed

Grouping has been added to the figure description.

Reviewer 3 Report

Review reports about

 Characterization of the Gut Microbiome in Healthy Dogs and 2 Dogs with Diabetes Mellitus    
1) A brief summary :

Gut microbiome of of 16 diabetic and 32 healthy dogs living in Hong Kong were explored with16SrRNA sequencing. After several bioinformatic analysis, this work reveals that Clostridioides difficile, Phocaeicola plebeius, Lacrimispora indolis and Butyricicoccus pullicaecorum were found to be enriched in diabetic dogs and that several carbohydrate degradation metabolic pathways were more abundant in diabetic dogs.

2) General concept comments.
Article:

Weakness : the work is very complete but remains difficult to understand, mainly because of the number of figures which are subdivided and difficult to read. Moreover, the elements of physiopathology are not always clear   Forces : The analysis methods are very complete

3). Specific comments referring to line numbers, tables or figures that
point out inaccuracies within the text or sentences that are unclear.

  Lines 27-28: the sentence “Unlike humans, canine glycemic control only relies on insulin supplemen-26 tation alongside dietary control in most cases » is unclear, not very explicit and not necessarily justified. Please explain.   Line 32 : why did you test 16 diabetic and 32 healthy dogs ?     Lines 58-61 : « Current treatment of canine DM mainly relies on diet modification (reduction in carbohydrates and calories intake) and exogenous insulin treatment to maintain blood glucose under the renal threshold for the greatest possible duration within a 24-hour timeframe [6]. » This sentence is not true : in fact, it has been shown that reducing carbohydrate intake improves the condition of diabetics but, as regards energy intake, it does not necessarily have to be reduced.   Line 68 : if you are talking about several diseases, it would be wise to cite several bibliographical sources     Lines 80/88 : the selection criteria for diabetic animals must be developed, especially since on several occasions the authors discuss type I diabetes mellitus. Also, how the dogs were selected to be sure that they have type I diabetes ?   Line 91 : « Rectal or stool swab samples were collected from dogs diagnosed with Diabetes Mellitus and healthy dogs at NPVAH ». The conditions of collection and their modalities should be developed (delay…).   Line 114 : please add a reference for Adonis procedure.   Lines 164-170 : Subfigure 2D should be presented in the text before 2E and 2F.   Figure 2 : The figure legend needs to be expanded, especially since the explanations under the 2D subfigure are written too small to read.   Lines 181-185 : please add the p-values in the text.   Figure 4 : Sub figures A, B, D are illegible. It is necessary to put the p-value on the under figure 4C.  

Figure 3: please add all the p-values ​​and the name of the bacteria in the sub figures 3B, 3C, 3D and 3E.

Line 203 : WY-621, please choose sucrose degredation or sucrose invertase and explain.

Page 9 : The figure has no number, title or legend. Parts A and B are difficult to read.

Figure 5 : This figure is too small to be understood, it is illegible. Perhaps it would be appropriate to put it in additional material

Lines 249-250 : This sentence should be in the results part, especially since the 2D figure is unreadable.

This central result should be more developed and better compared to the various other publications (16, 19, 35) and with the microbiota of healthy dogs.

Line 259 : please add in dogs

Lines 268-269 : why do you talk about type I diabetes in humans, when you do not explain how you only selected dogs with type I diabetes mellitus ?   Lines 286-289 : please better explan the link between such phyla and their ability to live in a carbohydrate-rich environnement.

Line 293 : do you think we can really talk about a cohort?

Lines 298-301 : At the end of the text, please be careful about the possible consequences of the greater presence of Cl. difficile in diabetics.

References :

- many journal names are abbreviated, it should not   Line 355 : remove 2018   Line 414 : what is « aos »

Author Response

Thank you for your valuable comments and pointing out the readiness of the manuscript. Since we have performed extensive bioinformatic analysis in this study and thus visualization is crucial in presenting our findings. We’ve revised the figures to make sure they’re in high resolution accordingly and please also find the point-to-point response in the main text in blue as follow: 

*Lines 27-28: the sentence “Unlike humans, canine glycemic control only relies on insulin supplemen-26 tation alongside dietary control in most cases » is unclear, not very explicit and not necessarily justified. Please explain.   

The sentence has been revised as “In clinical practice, canine glycemic control mainly involves the life-long supplementation of insulin and restricted carbohydrate intake, which jeopardizes diabetic dogs’ quality of life and increases difficulty in disease control.” In lines 25 – 27.

*Line 32 : why did you test 16 diabetic and 32 healthy dogs ?

We intended to recruit more diabetic dogs for comparison, but we could only recruit 16 diabetic dogs at the end of the study and thus applied non-parametric statistical tests to accommodate the uneven sample size. We will try to recruit more subjects in the future study to yield more representative results.

*Lines 58-61 : « Current treatment of canine DM mainly relies on diet modification (reduction in carbohydrates and calories intake) and exogenous insulin treatment to maintain blood glucose under the renal threshold for the greatest possible duration within a 24-hour timeframe [6]. » This sentence is not true : in fact, it has been shown that reducing carbohydrate intake improves the condition of diabetics but, as regards energy intake, it does not necessarily have to be reduced.   

Agreed. The wordings “calories intake” were removed from line 60.

*Line 68 : if you are talking about several diseases, it would be wise to cite several bibliographical sources

Four different studies on cancer, cardiovascular and gastrointestinal diseases, i.e. citations [16-19], were included in this sentence.

*Lines 80/88 : the selection criteria for diabetic animals must be developed, especially since on several occasions the authors discuss type I diabetes mellitus. Also, how the dogs were selected to be sure that they have type I diabetes ?

Unlike humans, there is no specific typing criteria of canine diabetes mellitus, and the selection criteria of diabetic subjects were clearly written in lines 89 - 90. We intended to mention human type I diabetes in both introduction and discussion sections for easier understanding because of the similarity of pathophysiology of diabetes mellitus between humans and canines.

*Line 91 : « Rectal or stool swab samples were collected from dogs diagnosed with Diabetes Mellitus and healthy dogs at NPVAH ». The conditions of collection and their modalities should be developed (delay…).

Sterile swabs were used for collection and the use of swabs have been added in lines 94 - 95.

*Line 114 : please add a reference for Adonis procedure. 

Citation [30] has been added for Adonis procedure as suggested.

*Lines 164-170 : Subfigure 2D should be presented in the text before 2E and 2F.   Figure 2 : The figure legend needs to be expanded, especially since the explanations under the 2D subfigure are written too small to read.

Figure 2 has been moved in the text before 2E and 2F. Also, the figure legend of 2D has been elaborated for clear illustration.

*Lines 181-185 : please add the p-values in the text.   

p-value have been added.

*Figure 4 : Sub figures A, B, D are illegible. It is necessary to put the p-value on the under figure 4C.  

p-value has been added to Figure 4C and Figure 4 has been replaced with larger size and higher resolution version. It should be readable now.

*Figure 3: please add all the p-values ​​and the name of the bacteria in the sub figures 3B, 3C, 3D and 3E.

All p-values have been indicated as *** in the subfigures and the key ***p<0.0001 is located at the bottom right of the subfigures. The names of bacteria are labelled in the Y-axis and description added in the figure legend for easier reading.

*Line 203 : WY-621, please choose sucrose degredation or sucrose invertase and explain.

Sucrose degradation pathway has been further elaborated in lines 212 - 213 as suggested.

*Page 9 : The figure has no number, title or legend. Parts A and B are difficult to read.

Figure 5 was accidentally separated into two pages thus made them difficult to read. The problem has been solved by splitting the legends.

*Figure 5 : This figure is too small to be understood, it is illegible. Perhaps it would be appropriate to put it in additional material

Figure 5 has been modified and a higher resolution version of figure has replaced the original one.

*Lines 249-250 : This sentence should be in the results part, especially since the 2D figure is unreadable. This central result should be more developed and better compared to the various other publications (16, 19, 35) and with the microbiota of healthy dogs.

Description of the distinct clustering in PCoA biplots have been added in lines 163 - 164. Since the subjects recruited in other publications (16,19,35) were not diabetes but cancer, it would be inappropriate for direct comparison in beta diversity. Instead, the predominant phyla were similar in our study with those in other publications (16,19,35) and thus included in the discussion section lines 265 - 268.

*Line 259 : please add in dogs

“in dogs” was added.

*Lines 268-269 : why do you talk about type I diabetes in humans, when you do not explain how you only selected dogs with type I diabetes mellitus ?

We intended to mention human type I diabetes mainly because the pathophysiology of diabetes mellitus between humans and canines are similar and it would be worthwhile for comparison and better understanding for readers.

*Lines 286-289 : please better explan the link between such phyla and their ability to live in a carbohydrate-rich environnement.

Since the casual relationship has not yet established for the outgrowth of microorganisms in carbohydrate-rich environment, our speculation on the underlying reasons have been discussed in lines 289 - 303.

*Line 293 : do you think we can really talk about a cohort?

The wording “cohort” has been replaced by “study population” for precision and consistency.

*Lines 298-301 : At the end of the text, please be careful about the possible consequences of the greater presence of Cl. difficile in diabetics.

The possible consequences of the enriched C. difficile in diabetic dogs have been discussed in full details (lines 274 – 288).

References :

- many journal names are abbreviated, it should not   

The citations were exported from citation reference software Mendeley with the style “Multidisciplinary Digital Publishing Institute”. In fact, the journal names should be abbreviated as instructed by the publisher, which can be found under “Back matter” in the publisher instruction page: https://www.mdpi.com/journal/animals/instructions

*Line 355 : remove 2018   

“Banfield State of Pet Health Report 2016” is the correct name of the publication, and it was published in year 2016.

*Line 414 : what is « aos »

The journal name has been updated to “Ann. Statist.”.

Reviewer 4 Report

The paper describes characterization of the gut microbiome of dogs with diabetes mellitus in Hong Kong, with comparison to healthy controls.

Canine gut microbiome is an important and 19 emerging field of veterinary research with promising potentials in facilitating disease diagnosis and 20

management.

Not sure how the microbiome would facilitate a diagnosis in DM in dogs.

potential application on clinical management. 23

I would delete this as I don’t think this is true.

Characterization of the Gut Microbiome in Healthy Dogs and 2 Dogs with Diabetes Mellitus

Gut should be replaced with fecal microbiome throughout the paper.

Through out the entire paper consistency is lacking in diabetes, diabetic, diabetes mellitus, DM, Diabetes Mellitus

Please pick one and stay consistent with the paper.

Diabetes Mellitus were enrolled from the 80

Dogs diagnosed with diabetes mellitus were selected 87

Current treatment of canine 58 DM mainly relies on diet modification (reduction in carbohydrates and calories intake) 59

I do not completely agree with this as this is more a feline thing than a canine thing.

3.1. Study Population 132

I do think, body score condition and maybe diet needs to incorporated into this study

Faith’s phylogenetic diversity (PD) was marginally significantly lower in the adult 148

I would delete either marginally as significant is significant.

elevated in the diabetic group (log 203 LDA = 2.919, p = 0.002), which may reveal the potential functional role of gut microbiota 204 in alleviating hyperglycaemia in the diabetic dogs. 205

Reference?

Canine DM, an aggressive and 239 difficult-to-manage chronic metabolic disease,

I find that the term aggressive is not suitable for describing diabetes mellitus.I would change this word.

In the present study, significant differential gut microbiota between diabetic and 246 healthy groups was observed

I would also stick with using fecal microbiota throughout the paper.

Thus, veterinarians should consider the gut 263 dysbiosis in clinical management of diabetic dogs to avoid gastroenteric complications, as 264 well as the enhanced susceptibility to C. difficile infection of canines with DM. 265

This is a stretch. Rephrase.

I do think it would interesting to discuss felines in brief in the context of small animal diabetes

·         Sci Rep . 2019 Mar 18;9(1):4822. doi: 10.1038/s41598-019-41195-0.

·         https://doi.org/10.1371/journal.pone.0108729

I would discuss the increased prevalence in the poodle.

'In a study of Swedish insurance data, poodles had an incidence risk of diabetes mellitus of 24 cases per 10,000 dog years at risk (95% Confidence Interval: 16-32 cases), whereas for the overall population it was 13 cases per 10,000 dog years at risk (Fall et al 2007). Comparing cases of diabetes mellitus in the Veterinary Medical Database in North America, poodles had a significantly increased risk of diabetes mellitus, compared to mixed breeds (Miniature poodle: odds ratio: 1.79, 95% CI: 1.55-2.06; Toy poodles: odds ratio: 1.29, 95% CI: 1.03-1.63; Guptill et al 2003). Similarly in the UK, Miniature poodles were found to be 2.31 times more likely to have diabetes mellitus than mixed breed dogs (95% CI: 1.14-4.68; Catchpole et al 2005.'

https://www.ufaw.org.uk/dogs/poodle---diabetes-mellitus

Minor concerns

Author Response

Thank you for your valuable comments. We’ve revised the main text accordingly and please find the point-to-point response in blue as follow: 

Canine gut microbiome is an important and 19 emerging field of veterinary research with promising potentials in facilitating disease diagnosis and 20 management.

Not sure how the microbiome would facilitate a diagnosis in DM in dogs.

We have elaborated this point in our discussion section with evidence found in this study.

potential application on clinical management. 23

I would delete this as I don’t think this is true.

The wordings have been changed to “implications to disease control”.

Characterization of the Gut Microbiome in Healthy Dogs and 2 Dogs with Diabetes Mellitus

Gut should be replaced with fecal microbiome throughout the paper.

We think that “Gut” would be more appropriate for describing our sample collection methodology as both rectal and faecal samples were involved in this study.

Through out the entire paper consistency is lacking in diabetes, diabetic, diabetes mellitus, DM, Diabetes Mellitus

Please pick one and stay consistent with the paper.

Diabetes Mellitus were enrolled from the 80

Dogs diagnosed with diabetes mellitus were selected 87

Diabetic is the adjective form for Diabetes Mellitus and the choice of using these two words depends on the situation. However, we agreed that diabetes mellitus should not be used and all “diabetes mellitus” and “DM” were replaced by “Diabetes Mellitus” throughout the manuscript.

Current treatment of canine 58 DM mainly relies on diet modification (reduction in carbohydrates and calories intake) 59

I do not completely agree with this as this is more a feline thing than a canine thing.

The current treatment of canine Diabetes Mellitus quoted from the 2018 report issued by the American Animal Hospital Association (reference 6), which is one of the largest accreditation organizations for veterinary hospitals.

3.1. Study Population 132

I do think, body score condition and maybe diet needs to incorporated into this study

It’s a good suggestion and we will consider for these two possible confounding factors in future studies.

Faith’s phylogenetic diversity (PD) was marginally significantly lower in the adult 148

I would delete either marginally as significant is significant.

The word “marginally” has been deleted.

elevated in the diabetic group (log 203 LDA = 2.919, p = 0.002), which may reveal the potential functional role of gut microbiota 204 in alleviating hyperglycaemia in the diabetic dogs. 205

Reference?

Elaboration of this point have been discussed with references. The sentence has been modified to refer the readers to see the discussion section.

Canine DM, an aggressive and 239 difficult-to-manage chronic metabolic disease,

I find that the term aggressive is not suitable for describing diabetes mellitus.I would change this word.

The word “aggressive” has been changed to "disastrous”.

In the present study, significant differential gut microbiota between diabetic and 246 healthy groups was observed

I would also stick with using fecal microbiota throughout the paper.

We think that “Gut” would be more appropriate for describing our sample collection methodology as both rectal and faecal samples were involved in this study.

Thus, veterinarians should consider the gut 263 dysbiosis in clinical management of diabetic dogs to avoid gastroenteric complications, as 264 well as the enhanced susceptibility to C. difficile infection of canines with DM. 265

This is a stretch. Rephrase.

The sentence has been rephrased into “Veterinarians should consider the gut dysbiosis in clinical management of diabetic dogs and gut microbiome testing may serve as a handy tool to reveal enhanced susceptibility to C. difficile infection of canines with Diabetes Mellitus.”.

I do think it would interesting to discuss felines in brief in the context of small animal diabetes

  • Sci Rep . 2019 Mar 18;9(1):4822. doi: 10.1038/s41598-019-41195-0.
  • https://doi.org/10.1371/journal.pone.0108729

We would concentrate on dogs in this manuscript as this is the main context of the study. However, we will take into consideration to study feline Diabetes Mellitus in the future.

I would discuss the increased prevalence in the poodle.

'In a study of Swedish insurance data, poodles had an incidence risk of diabetes mellitus of 24 cases per 10,000 dog years at risk (95% Confidence Interval: 16-32 cases), whereas for the overall population it was 13 cases per 10,000 dog years at risk (Fall et al 2007). Comparing cases of diabetes mellitus in the Veterinary Medical Database in North America, poodles had a significantly increased risk of diabetes mellitus, compared to mixed breeds (Miniature poodle: odds ratio: 1.79, 95% CI: 1.55-2.06; Toy poodles: odds ratio: 1.29, 95% CI: 1.03-1.63; Guptill et al 2003). Similarly in the UK, Miniature poodles were found to be 2.31 times more likely to have diabetes mellitus than mixed breed dogs (95% CI: 1.14-4.68; Catchpole et al 2005.'

https://www.ufaw.org.uk/dogs/poodle---diabetes-mellitus

The increased prevalence in poodles have been mentioned in line 313 - 315 and the references have been updated.